# Comparison of DNA Extraction Methods for the Direct Quantification of Bacteria from Water Using Quantitative Real-Time PCR

**Kousar Banu Hoorzook *** and **Tobias George Barnard**

Water and Health Research Centre, Faculty of Health Sciences, University of Johannesburg,
Johannesburg 2028, South Africa
*  Correspondence: kousaro@uj.ac.za; Tel.: +27-(0)1-1559-6567; Fax: +27-(0)1-1559-6329

**Abstract:** Isolating DNA from bacterial cells concentrated directly from water samples allows the analysis of the DNA with a range of molecular biology techniques. The aim was to develop a cost-effective method to concentrate bacterial cells directly from water for DNA extraction and PCR amplification. A modified in-house guanidinium thiocyanate DNA extraction method was compared to four commercial kits (two repeats performed in triplicate) from 10-fold serially diluted bacterial cells and used to construct standard curves using quantitative real-time PCR (q-PCR). The in-house DNA extraction method-constructed qPCR standard curves showed similar results with determination coefficient ($R^2$) of 0.99 and 0.99 and of slopes $-3.48$ and $-3.65$). The $R^2$ and slope for Water Master™ DNA purification kit ($R^2$ 0.34, 0.73; slope $-5.73$, $-4.45$); Ultra Clean™ Water DNA isolation kit ($R^2$ 0.97, 0.28; slope $-3.89$, $-8.84$); Aquadien™ kit ($R^2$ 0.98, 0.77; slope $-3.59$, $-5.94$) and Metagenomic DNA isolation kit ($R^2$ 0.65, 0.77; slope $-3.83$, $-4.89$) showed higher variability than the in-house DNA extraction method. The results showed that the in-house DNA extraction method is a viable cost-effective alternative with good DNA recovery and repeatable and reproducible results. A limitation of the study is the limited number of repeats, due to cost implication of the commercial kits.

**Keywords:** commercial water-testing kits; DNA extraction; *E. coli*; q-PCR





## 1. Introduction

There are numerous methodologies for the detection of specific bacteria from environmental samples that include microbiology and molecular biology [1]. Conventional culture-based methods have limitations from both quantitative and qualitative points of view. When applied to the detection of pathogens in the environment, culture-based protocols may be inaccurate due to the selective nature of the media, which require the use of specific media and specific culture conditions for each sought microorganism [2]. Further confirmations are required after culturing to distinguish the diarrhoeagenic *Escherichia coli* (DEC) from commensal *E. coli* (ComEC) [3]. More accurate detection, obviating the need for cell culture can be achieved by molecular biology techniques [4,5]. Molecular biology analysis can offer various advantages over cultural methods, including detection of a wider range of target organisms, greater sensitivity, and specificity. Molecular methods have traditionally been performed on single isolates and from enrichments. Enrichments provide higher bacterial counts but only indicate viable bacteria, cannot estimate counts, and only get an idea of presence and absence. Since viable but non-culturable bacteria cannot be isolated by standard culture-based methods, the simplest way to overcome this is to isolate DNA from bacterial cells concentrated directly from the water samples. This isolated DNA can then be used as a template for polymerase chain reaction (PCR), thus circumventing the need for culturability [6,7]. The purity, yield, and quality of DNA extracted from a heterogeneous material are a key issue in the sensitivity and usefulness of further analysis, such as PCR analysis for infectious pathogens [8]. Time constraints may

make traditional phenol–chloroform extraction of bacterial DNA impractical as additional clean-up procedures may be required to remove carry-over phenols, which inhibit PCR reactions [9,10]. Other methods have been developed, usually involving the lysis of bacteria and subsequent binding of released DNA onto a solid matrix, followed by washing and elution of the relevant components. An example of these methods is the [11] method, with the addition of guanidium isothiocyanate for lysis of bacterial cells, with a silica matrix to hold DNA allowing the wash removal of proteins and final elution of DNA. Additionally, commercial kits have been developed for water samples and have gained preference among researchers. Commercial DNA isolation kits have the advantage of using only a small amount of chemicals and of achieving results more rapidly. However, they have noteworthy disadvantages such as high costs, non-repeatability of the DNA yield, and purity and time constraints in processing samples [1,8]. The relative efficiency and efficacy of these extraction methods has not been fully explored. In addition, the problem of importing commercial kits during the COVID-19 pandemic and lockdown when it takes months to receive consumables would be resolved by using inexpensive and in-house methods of DNA extraction.

The aim was to develop a cost-effective method to concentrate the bacterial cells directly from water samples followed by DNA extraction from the cells. This method was compared to commercially available water-testing DNA extraction kits. The key issues evaluated in each method were concentration, sensitivity, extraction efficiency, repeatability, and reproducibility.

## 2. Materials and Methodology

### 2.1. Growth and Maintenance of Bacterial Strains

The commensal non-pathogenic *E. coli* (ComEC) and pathogenic Entero-haemorrhagic *E. coli* (EHEC) strains (ESCCO 21) were cultured from a frozen stock culture on Plate Count Agar (PCA) (Oxoid, Basingstoke, UK) and incubated under aerobic conditions at 37 °C for 16 h. Single colonies were enriched in 100 mL nutrient broth and incubated under aerobic conditions at 37 °C for 16 h with rotation at 200 rpm.

EHEC cells in four 50 mL Eppendorf tubes were diluted with water to obtain an $OD_{600nm} = 1.0$ and then centrifuged for 2 min at $13,000 \times g$ rpm. The supernatant was discarded, and the pellets washed twice and then re-suspended with 500 uL of water. The cell pellets in the tubes were pooled and 500 uL from the total volume was used for the $OD_{600nm}$ reading. From the remaining suspension, 900 uL was used for the positive control (PTC) and the rest was used making the ten-fold dilutions in triplicate for membrane filtration.

### 2.2. Comparison of Optimised DNA Extraction Method with Commercial Water-Testing Kits

2.2.1. DNA Extraction

Buffer Preparations

The preparation of the celite, lysis buffer, washing buffer, and spin columns used for the optimized in-house DNA extraction method is as follows [12]:

Celite

Celite (Sigma Aldrich, St. Louis, MO, USA) was prepared by suspending 10 g in 50 mL distilled water and adding 500 μL hydrochloric acid (HCl) (32% $w/v$) to the solution. Thereafter it was sterilized for 15 min at 121 °C and the bottle wrapped in aluminum foil (celite solution is sensitive to light) and refrigerated at −20 °C (stable for 3 weeks at room temperature).

Lysis buffer

Lysis buffer was prepared by dissolving 120 g guanidinium thiocyanate (GuSCN) (G6639) in 100 mL of 0.1 M hydroxymethyl amino methane-hydrochloric acid (Tris-HCl) (pH 6.4) in a 500 mL Schott bottle. The bottle was heated to 60 °C to dissolve the GuSCN. If not heated, the GuSCN will not dissolve. After heating, 22 mL of a 0.2 methylenediamine tetra-acetate (EDTA) (pH 8.0) with 2.6 mL triton X-100 (Sigma Aldrich, St. Louis, MO,

USA) solution was added to the suspension. The suspension was mixed and dispensed into 50 mL Eppendorf tubes and 0.5 mL of celite suspension was added to remove any contaminating DNA from the buffer. The final solution was left to stand for at least 1 h at room temperature with sporadic mixing. The celite was pelleted by centrifugation at 3000 rpm for 10 min (NeoFuge-15R, Heal Force, Vacutec®, Dresden, Germany) and the supernatant was transferred into sterile 50 mL Eppendorf tubes wrapped in aluminum foil (sensitivity towards light) (stable 3 weeks at room temperature).

Washing buffer

Washing buffer was made up by dissolving 120 g GuSCN and 100 mL of 0.1 M Tris-HCl (pH 6.4) in a 500 m Schott bottle, heated to 60 °C to dissolve the GuSCN, and dispensed into 50 mL Eppendorf tubes. Thereafter, 0.5 mL celite suspension was added to each tube to remove contaminating DNA from the buffer as described above. Washing ethanol A 70% (*v/v*) ethanol solution was prepared with sterile distilled water.

Preparation of spin columns

The cap off 0.5 mL Eppendorf tubes were cut leaving the small tail behind. Several holes were made in the bottom of the tube with a red-hot needle. Important to note, the holes should not be too small or too big, otherwise the filters will get blocked or the celite solution will run out of the holes and not be retained. Silica membranes were cut from GF/F filter paper (Cat log no. 1825-037; Merck SA, Lethabong, South Africa) using 5 mm punch. Two membranes were tightly inserted into an Eppendorf tube (Figure 1). The tubes are sterilized in glass jars for 15 min at 121 °C.

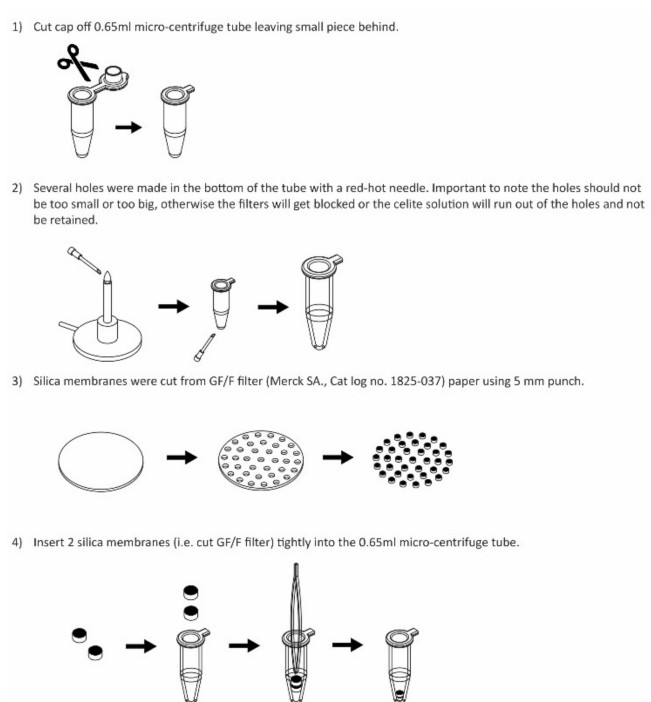

**Figure 1.** Process in making spin-columns in the laboratory.

The optimized in-house DNA extraction method was based on the method reported by [13], who used a modification of the Boom et al. [11] and Borodina et al. [14] protocols. For the optimized in-house DNA extraction method, the bacterial cells were grown as described in Section 2.1, the bacterial cells were filtered onto polycarbonate (Poly) membrane using the standard membrane filtration technique. The protocol was followed as: the overnight bacterial cells were filtered onto the Poly membrane in triplicate and the membrane was placed into 4 mL Ogreiner Bio-1 cryovial (Lasec, Cape Town, South Africa) with sterile forceps. The filters were vortexed for 2 min with 2.5 mL sterile distilled water. The suspension was transferred into 2 mL Eppendorf tubes, centrifuged at 13,000 rpm

for 10 min and the supernatant discarded. Lysis buffer, 1000 μL was added to the pellets and negative control tube. The samples were incubated at 70 °C for 10 min. Thereafter, 200 μL 100% ethanol was added to each tube. Samples were mixed with gentle swirling and incubated at room temperature for 10 min. Approximately 400 μL of this suspension was transferred into prepared spin columns and centrifuged at 13,000 rpm for 1 min to pellet the celite. The collection tubes were emptied into designated DNA waste containers and the above step was repeated twice. This was followed by two more wash steps using 400 uL of wash buffer followed by centrifugation at 13,000× $g$ rpm for 30 s. The collection tubes were emptied into the designated DNA waste container and the above step was repeated. Thereafter, 400 uL of 70% ($v/v$) ethanol solution was added to each spin column and centrifuged at 13,000× $g$ rpm for 30 s. The collection tubes were emptied into a designated DNA waste container and the above step was repeated. After discarding the supernatant of the second repeat, the pellet was dried by centrifuging the tubes at 13,000 rpm for 3 min. The spin columns were placed into new labelled 1.5 mL Eppendorf tubes and 100 uL of elution buffer was added to each spin column and incubated at 37 °C for 2 min. After incubation, the samples were centrifuged at 13,000× $g$ rpm for 2 min during which the DNA was eluted into the Eppendorf tubes.

Thereafter, four commercial water-testing kits were compared with the in-house DNA extraction method. The commercial water-testing kits used were the Aquadien™ kit (Cat log no. 3578121; Bio-Rad; Johannesburg, South Africa); Ultra Clean™ water DNA isolation kit [Cat log no. 14800-25; Optima Scientific PTY (LTD); Cape Town, South Africa]; Water Master™ DNA purification kit (Cat log no. MGD08420; Separations; Johannesburg, South Africa) and Metagenomic DNA isolation kit for water (Cat log no. MGD08420; Separations; Johannesburg, South Africa). These kits were selected due to their availability, cost, ease of use, and popularity. The DNA extractions were performed in triplicate and repeated on two separate days to test for repeatability and reproducibility. More repeats could not be performed due to the cost of each kit and limited budget.

The extracted DNA was used in the q-PCR reaction and was quantified in ng/μL using the Qubit™ fluorometer (Invitrogen, Waltham, MA, USA). The [DNA] values were converted into copies/μL using Equation (1) [15] and used as the starting template concentration in the q-PCR analysis.

$$\text{Number of initial copies}/\mu L = \left( x_{(g/L)} \times 6.022 \times 10^{23} \left(\text{mol}^{-1}\right) \middle/ \left(4.7 \times 10^6 \text{bp}\right) \times \left(660\,\text{g} \times \text{mol}^{-1}\right) \right) \quad (1)$$

where $x$ (g/L) is the DNA concentration, $6.022 \times 10^{23}$ (mol$^{-1}$) is the Avogadro constant, $4.7 \times 10^6$ bp is the size of the complete *E. coli* genome, and $660\,\text{g} \times \text{mol}^{-1}$ is an assumption of the average weight of the base pair.

### 2.2.2. Quantitative Real-Time Polymerase Chain Reaction (q-PCR)

q-PCR reactions were performed in a Corbett Research Thermal cycler (Celtic Molecular Diagnostic (PTY) LTD, Cape Town, South Africa) in a total volume of 20 μL. A HotStart PCR kit (Qiagen®, Hilden, Germany) was used for the q-PCR protocols. Each reaction consisted of 1X 2 μL Qiagen® PCR buffer mix; 0.1 μL of 5 units/μL HotStart Taq® DNA polymerase and 0.6 μL of 400 μM dNTP mix; 1 μL of 3 μM probe (Table 1); 2 μL of 3 Mm Mg$^{2+}$; 1 μL of each 5 μM forward and reverse primer (Table 1); 3 μL of sample DNA and 9.3 μL PCR grade water. For two-point dilution on the standard curve $10^6$ and $10^5$ DNA of referenced ComEC were included with all q-PCR reactions. The PCR reactions were subjected to a 2-step q-PCR protocol. For *gadAB* protocol, an initial enzyme activation step at 95 °C for 15 min was followed by 35 cycles of a 94 °C denaturation for 15 s and a 55 °C elongation step for 60 s. Negative control reaction mixtures contained sterile PCR grade water in place of template DNA.

**Table 1.** Primers and probe used in the q-PCR reaction.

| Primers | Sequence (5′-3′) | Amplicon Size | Patho-Type | Reference |
|---------|------------------|---------------|------------|-----------|
| *gadAB*-F *gadAB*-R P | GCGGAAGTCCCAGACGATATCC GCTACACGTACAGCTACAGCTA r-CGGTGRCMGGAMGCRA-q | 670 bp | All *E. coli* strains | Designed by Sophi Breniere (Sigma France) [16] |

q—Black Hole Quencher (BHQ-1); r—6-carboxyflourescein (FAM).

### 2.3. Statistical Analysis

The extracted DNA was quantified in ng/μL using the Qubit$^{TM}$ fluorometer (Invitrogen, Waltham, MA, USA). This was converted into copies/μL [15] and used as the starting template concentration (input DNA) in the qPCR analysis. Statistical analysis was performed using Graphpad Prism$^{®}$ 9 and IBM SPSS statistics 23. To facilitate further analysis via one-way ANOVA and post-hoc tests, Shapiro–Wilk measure of normality and Levene test of homogeneity of variances were assessed. The non-parametric test was used to check for any contradictions to the parametric tests because there were less than 30 observations per sample. The Kruskal–Wallis test was used to see if there are significant differences in the mean scores on the dependent variables across the groups. This test is an alternative to one-way ANOVA. The Mann–Whitney U test was used to ascertain where these differences lie. This test is an alternative to the Post-hoc test.

Using the in-house DNA extraction method and the commercial water-testing kits, we calculated the z-scores. Most laboratories utilize proficiency testing (PT) to monitor and optimize the quality of the routine analytical measurements. The organizers carry out statistical analysis of all the data and provide participants with a "score" that allows them to judge their performance in a particular round. The most common scoring system is the z-score (Equation (2)):

$$Z = \frac{X - Xa}{\sigma p}. \tag{2}$$

## 3. Results and Discussion

*Validation of the In-House DNA Extraction Method against Commercial Water-Testing Kits*

The in-house DNA extraction method was compared with commercially available water-testing kits in terms of effective bacterial cell concentration, DNA extraction efficiency, repeatability, and reproducibility. Therefore, in this study, the in-house DNA extraction method became the basis for different commercially available water-testing kits. The commercial water-testing kits used were the Aquadien$^{TM}$ kit (Bio-Rad, Johannesburg, South Africa); Ultra Clean$^{TM}$ water DNA isolation kit (Optima Scientific PTY (LTD), Cape Town, South Africa); Water Master$^{TM}$ DNA purification kit (Separations, Johannesburg, South Africa), and Metagenomic DNA isolation kit for water (Separations, Johannesburg, South Africa). These commercial water-testing kits were chosen based on availability, cost, ease of use, and popularity.

In these experiments, the bacterial cells first were diluted ten-fold before DNA extraction and q-PCR analysis was performed. The starting DNA concentrations for each commercial water testing kit added in the q-PCR analysis are indicated in Table 2. The optimized *gadAB* standard curve was selected to compare the optimized DNA extraction method and commercial water testing kits. *Gad* encodes for glutamate decarboxylase and *gadAB* gene is prevalent in all *E. coli* both pathogenic and non-pathogenic [16,17]. The optimized *gadAB* standard curve was imported to measure copies for the unknown samples using the Corbett Research Thermal cycler (Qiagen$^{®}$, Hilden, Germany) machine. The dilutions of $10^6$ and $10^5$ DNA of referenced EHEC or ComEC was included with all q-PCR reactions as a standard for quantification and not the $10^4$ to $10^1$ dilutions, to avoid human pipetting errors or DNA attaching to the tube walls (Figure 2a,b).

**Table 2.** Starting template DNA concentration added for the q-PCR standard curves.

| Commercial Kits | Initial Concentration (ng/µL) | Copies/3 µL |
|---|---|---|
| In-house DNA extraction | 4.93 | $1.9 \times 10^6$ |
| Aquadien$^{TM}$ kit | 5.4 | $2.7 \times 10^6$ |
| Ultra Clean$^{TM}$ water DNA isolation kit | 1.6 | $7.3 \times 10^5$ |
| Water Master$^{TM}$ DNA purification kit | 4.79 | $1.9 \times 10^5$ |
| Metagenomic DNA isolation kit | 1.5 | $5.9 \times 10^4$ |

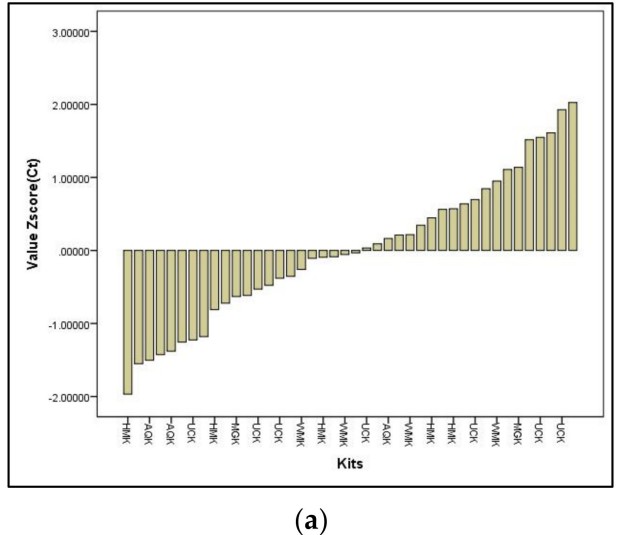

(**a**)

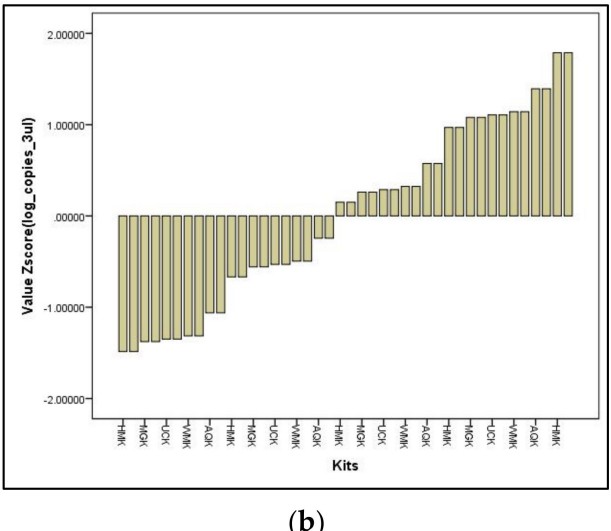

(**b**)

**Figure 2.** (**a**) Z value scores for $C_q$ for in-house DNA extraction and the commercial water-testing kits; (**b**) Z value scores for log copies/3 µL for in-house DNA extraction and the commercial water-testing kits.

The standard curves were plotted to test the relationship between the input DNA concentrations determined to calculate the DNA concentrations from the Corbett Research Thermal cycler software. The concept is that if the DNA concentration was determined correctly i.e., no influence from the extraction method, a perfect linear relationship is obtained. The PCR efficiency was estimated through the linear regression of the dilution curve. The determination coefficient for the in-house DNA extraction method was $R^2$ = 0.99 and $R^2$ = 0.99 for two repeats, indicating there was no influence from the DNA extraction method (Table 3, Figure 3a). The higher the $R^2$ or closest to 1 indicates more robust models [18] and is also a measure of the accuracy of the dilutions and precision of pipetting [19]. The determination coefficient and slope for the commercial water-testing standard curves are indicated in Table 3 (Figure 3b–e). The in-house DNA extraction method was similar to the Aquadien$^{TM}$ kit (that also uses Poly membrane) ($R^2$ = 0.92), and not to the other commercial water-testing kits whereby $R^2$ ranged from 0.24 to 0.98. The slope for two repeats of the optimized DNA extraction standard curve were −3.48 and −3.65 (Table 3). These values were 94 and 88% close to that of a PCR with an efficiency of approximately 100% (−3.30). Generally, PCR reactions do not reach 100% efficiency due to experimental limitations [20]. The y-intercept indicates the sensitivity of the assay and how accurately the DNA template has been quantified [19]. The y-intercept differences for the in-house DNA extraction method are 31.5 and 34.3 and the commercial water-testing kits y-intercept is indicated in Table 3. To note a limiting factor is that an extraneous DNA e.g., salmon sperm DNA was not included to each sample in this DNA extraction process to accurately measure the extraction efficiency.

**Table 3.** Linear Regression summary of the in-house DNA extraction method and commercial water-testing kits.

|  |  | In-House DNA Extraction | Ultra Clean™ Water | Aquadien™ | Metagen-Omic | Water Master™ |
|---|---|---|---|---|---|---|
| Input DNA | $R^2$ | 0.99 | 0.98 | 0.92 | 0.65 | 0.34 |
|  | Slope | −3.48 to −3.65 | −3.9 | −3.6 to −2.1 | −2.5 | −5.7 |
|  | Y-intercept | 31.5 to 34 | 32 to 39 | 32 to 33 | 28 to 33 | 36 |
| Input DNA vs. Calc. DNA | $R^2$ | 0.99 | 0.43 | 0.92 | 0.81 | 0.24 |
|  | Slope | 1.1 | 0.49 to 1.72 | 0.99 | 0.8 | 0.5 |
|  | Y-intercept | −0.41 | −2.2 to 1.8 | −0.36 to 0.35 | −0.6 to 0.8 | −0.59 to 2.1 |

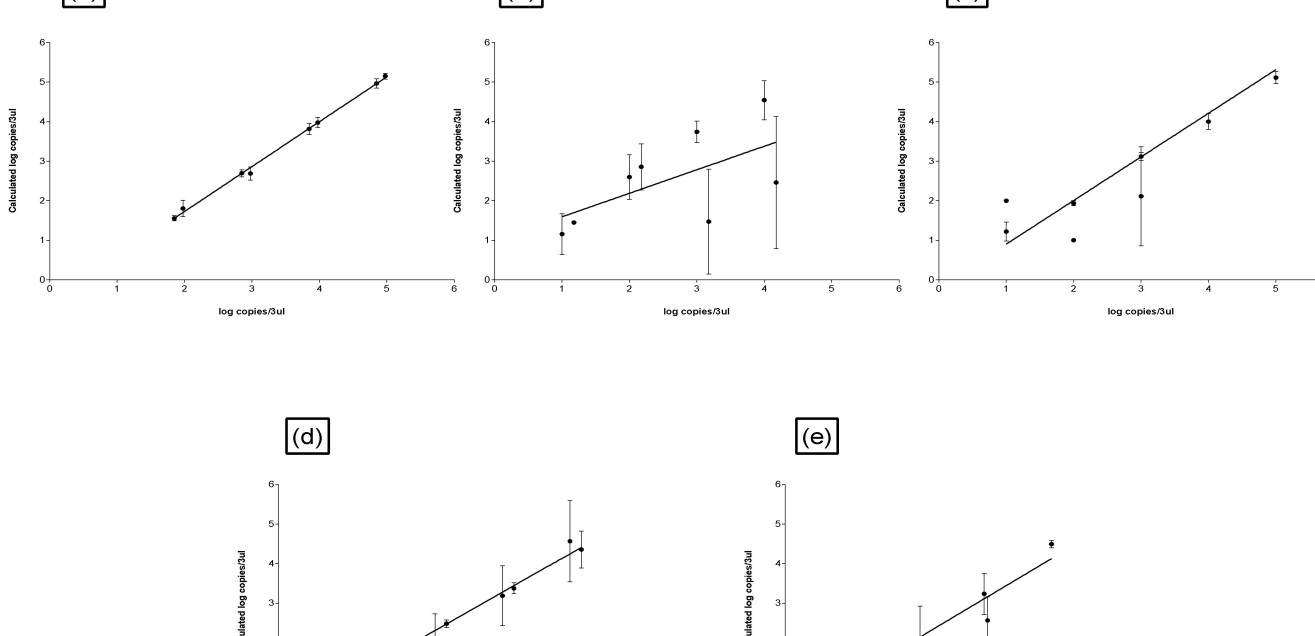

**Figure 3.** Input DNA versus calculated DNA from the Corbett Research Thermal Cycler software for (**a**) in-house DNA extraction method, (**b**) Water Master™ DNA purification kit (Separations, Johannesburg, South Africa), (**c**) Ultra Clean™ water DNA isolation kit (Optima Scientific PTY (LTD), Cape Town, South Africa), (**d**) Aquadien™ kit (Bio-Rad, Johannesburg, South Africa), (**e**) Metagenomic DNA isolation water kit (Separations, Johannesburg, South Africa); performed in triplicate with two repeats.

Statistical analysis indicated that the $R^2$ values did not show a significant difference between the in-house DNA extraction method and the three commercial DNA extraction water-testing kits ($p \geq 0.05$), except for the Water Master™ DNA purification kit ($p = 0.021$) (Table 4). Adams et al. [19] stated that data should be presented in a manner that allow the reader to observe the amount of variation inherent to the experiment, for example mean standard deviation (SD) and confidence intervals (Table 4). In biological systems, there is variation and experimental imprecision, and some statistics can reveal differences that are not otherwise discernible [19]. The coefficient of variation (CV) was used to measure intra-assay reproducibility from well to well and inter-assay variation from assay to assay, therefore, the smaller the CV the better the assay. SD and coefficient of variation (CV) for the $R^2$ were therefore included for these analyses. The SD and CV respectively for the

$R^2$ for the optimized DNA extraction method were lower (0.0035% and 0%) compared to Water Master[TM] DNA purification kit (0.1% and 22%); Ultra Clean[TM] Water DNA isolation kit (0.5% and 90%); Aquadien[TM] kit (0.1% and 12%); Metagenomic DNA isolation kit (0.8% and 12%). As reported by Adams et al. [19], SD is a good indicator of how much variability there is in the data.

**Table 4.** Statistical comparison between in-house DNA extraction method versus commercial water-testing kits.

| Treatment Name | Mean | Std. Dev | Std. Error | *p*-Value | CV |
|---|---|---|---|---|---|
| In-house DNA extraction | 0.991 | 0.00354 | | | 0% |
| Ultra Clean[TM] kit | 0.598 | 0.534 | | | 89.57% |
| In-house DNA extraction vs. Ultra Clean[TM] kit | 0.393 | 0.538 | 2.228 | 0.159 ** | |
| Aquadien[TM] kit | 0.904 | 0.105 | | | 11.72% |
| In-house DNA extraction vs. Aquadien[TM] kit | 0.0868 | 0.108 | −2.321 | 0.142 ** | |
| Water Master[TM] kit | 0.64 | 0.147 | | | 22.10% |
| In-house DNA extraction vs. Water Master[TM] kit | 0.351 | 0.143 | 3.663 | 0.021 * | |
| Metagenomic kit | 0.707 | 0.0823 | | | 11.95% |
| In-house DNA extraction vs. Metagenomic kit | 0.284 | 0.0788 | −1.036 | 0.0512 ** | |

NB: $p \geq 0.05$ non-statistically different **; $p \leq 0.05$ statistically different *.

The most common scoring system laboratories utilize for PT is the z-score to carry out statistical analysis of all the data [21]. For this study, Equation (2) (Statistical analysis) was adapted and "various laboratories" was substituted with the optimized DNA extraction method and the commercial water-testing kits. Whereby, $X$ is the result of each DNA extraction method, $Xa$ is each DNA extraction method's mean, and $\sigma p$ is the standard deviation of each DNA extraction method. z-scores are typically interpreted as the follows:

$Z \leq 2$ Satisfactory performance
$2 < Z \leq 3$ Questionable performance
$Z > 3$ Unsatisfactory performance

For these tests, the performance of the in-house DNA extraction method and commercial water-testing kits were satisfactory as they received the scores $Z \leq 2$ for both the $C_q$ and log copies/3 μL results.

However, the results indicated variability in repeatability, reproducibility, and sensitivity across the four commercial water-testing kits. In contrast, the in-house DNA extraction method allows for repeatability, reproducibility, and sensitivity. The comparison can be seen in Figure 3a–e, which show the input DNA versus the calculated DNA obtained from the Corbett Research Thermal cycler software for the in-house DNA extraction method and the commercial water-testing kits. This variation in repeatability, reproducibility and sensitivity is further demonstrated by the consistency of cells detected between the in-house DNA extraction method (613 cells/100 μL) and Aquadien[TM] (675 cells/100 μL) and the inconsistency between the remaining commercial water-testing kits (90, 39 and 1,911,422 cells/100 μL). This may be because the DNA binding system for the commercial water-testing kits is limited to adsorption into a silica membrane, whereas, the in-house DNA extraction method uses silica particles, in the presence of the chaotropic salt guanidium thiocyanate as well as the filter membrane in the spin column to capture the DNA [11].

Furthermore, the in-house DNA extraction method also proves to be comparable to the four commercial water-testing kits with respect to the upper and lower copy number limit (Figure 4), owing to good repeatability and reproducibility.

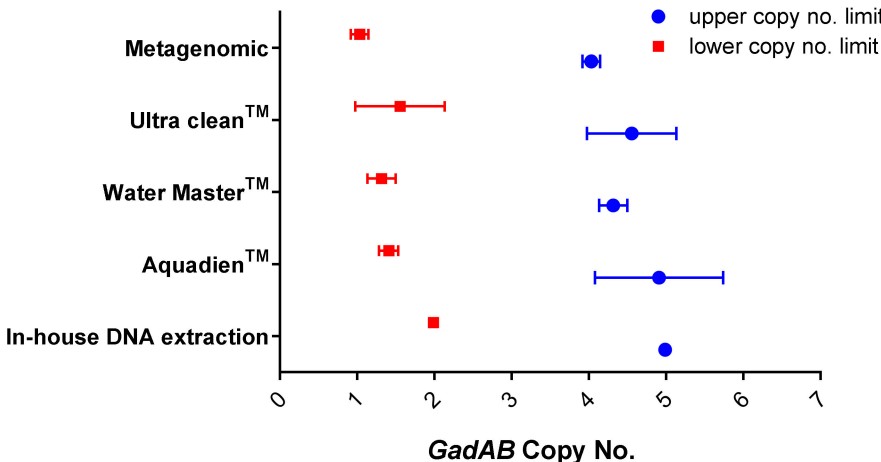

**Figure 4.** Comparing the in-house DNA extraction method and commercial DNA extraction water-testing kits upper and lower copy number limitations.

The most important thing to remember is that the DNA extraction methods differ a great deal, despite each experiment starting with a constant $OD_{600nm}$ of 1.0. There is an explanation for these differences in Table 5, where elution volume is different between methods, therefore diluting or concentrating DNA. It is interesting to note that, even though input DNA concentration varies, a certain $C_q$ relates to a DNA concentration i.e., for $C_q$, 18 to 20 only ~9310 copies/3 μL of DNA were detected for the in-house DNA extraction method and the commercial water-testing DNA extraction kits.

From the results obtained, two questions arose i.e.,

(1) Is the binding and loading capacity of the in-house DNA extraction method (Figure 5) responsible for a low copy number? The binding and loading capacity of the optimized DNA extraction method indicates that if you pool together concentrated *E. coli* cells for 0.1 or even 2.5 mL, the q-PCR $C_q$ values remain in the region of 12.5 and 16.5, even though there is an exponential increase in the concentration of *E. coli* cells (Figure 5).

(2) Why is there no substantial difference in the q-PCR values? It could be due to variables or limiting factors in the PCR reaction. Typically, a PCR reaction begins exponentially, then enters a quasi-linear phase, then plateaus. Several factors have been presumed to contribute to this plateau: (1) utilization of substrates (dNTPs and primer concentration), (2) thermal inactivation and limiting concentration of DNA polymerase, (3) the effect of DNA concentration as well as the effect of background DNA [22,23].

It was also encouraging to discover that the cost of performing the in-house DNA extraction method was much more cost-effective than using commercially available water-testing kits (Table 5). There are some of these kits that are not readily available in South Africa, contributing to an increase in cost due to import tariffs, shipping fees, and time, which takes approximately 4–6 weeks. It was found that the least expensive commercial water testing kit costs three times as much as the in-house DNA extraction method. These kits did not include the costs of the membrane used, which had to be bought separately. The in-house DNA extraction method price includes the complete process. Using the in-house DNA extraction method, the process is completed faster than using commercial water-testing kits. As shown in Table 5, the elution volumes of each kit differ, which obviously affects the overall sensitivity of the system. However, the evaluation was designed to compare sensitivities, repeatability, and reproducibility of the commercial water-testing kit when used as specified by the manufacturers for optimal performance and no effort was

made to standardize output volumes. Choosing a DNA extraction method also requires consideration of the additional reagents and special equipment required to complete the DNA extraction process which are not included in the kit, for example the Ultra Clean$^{TM}$ Water DNA isolation kit requires a 50 mL refrigerated centrifuge (Table 5). Akkurt [8] reported several disadvantages in the use of commercial DNA extraction kits, such as high costs, non-repeatability of the DNA yield and purity, and time constraints in processing samples. The relative efficiency and effectiveness of these extraction methods have not been fully explored.

**Table 5.** Cost comparison of in-house DNA extraction method and commercial DNA extraction water-testing kits, conversion at $15.

| Name | Cost per Kit | Cost/ Reaction | Volume Eluted (uL) | Time Taken for 24 Samples | Additional Equipment/Reagents (Not Supplied) | Cautionary for Method |
|---|---|---|---|---|---|---|
| Ultra Clean$^{TM}$ water DNA isolation kit (0.45 μm) | ~R18895 plus ~R13216 water filters for 25 reactions, excl. VAT | R520 | 3000 | ~5 h | - Bench centrifuge for 15 and 50 mL tubes<br>- Pipette from 50 μL to 10 mL<br>- Vacuum filtration system | • Make sure the 15 mL bead solution screw cap tubes rotate freely in your centrifuge without rubbing. Do not spin the bead tubes in excess of 6000 rpm.<br>• Your final volume will be 3 mL. If this is too diluted, can concentrate by adding NaCl, 100% ice-cold ethanol, mix and centrifuge at $2500 \times g$ for 20 min. Dry residual ethanol in a speed vac, desiccator, or ambient air. Re-suspend precipitated DNA in the desired volume. |
| Aquadien$^{TM}$ kit (Discontinued) | ~R12834 for 100 reactions excl. VAT | R150 | 100 | ~4 h | - Filtration apparatus<br>- Class II safety cabinet<br>- Water bath<br>- Vortex<br>- Bench centrifuge for 1.5–2.0 and 4.5 mL tubes<br>- Magnetic stir plate<br>- DNA free water<br>- 5% Sodium hypochlorite solution<br>- 70% Alcohol | • In order to keep the resin in suspension and collect it, pipette R1 must be stirred at medium speed on a stir plate. Use a pipette tip with a wide opening (e.g., use a 200 μL to 1000 μL pipette with the corresponding tip.<br>• The raw PCR result should be multiplied by 36 to obtain the final quantity of bacteria contained in the initial water sample, expressed in genomic unit (GU) per water sample liter. If the filtered water volume is different from 1 L, take it into account in these calculations. |
| Metagenomic DNA isolation kits for water | ~R3900 for 20 reactions excl. VAT | R170 | 50 | ~4 h | - Pre-sterilized 0.45-micron filter<br>- Micro-cloth filtration material or sterile cheesecloth<br>- 1.7 mL micro-centrifuge tubes<br>- Tween® 20<br>- Isopropanol<br>- 70% Ethanol | • After extractions do an end repair followed by ethanol precipitation to clean up the sample. |
| Water Master$^{TM}$ DNA purification kit (Discontinued) | ~R3410 for 20 reactions, excl. VAT | R205 | 60 | ~4 h | - Cheesecloth<br>- pre-sterilized 0.45-micron filter<br>- 70% Ethanol<br>- 50 mL conical tube<br>- 1.5 mL centrifuge tubes<br>- Micro-centrifuge<br>- Vortex<br>- Heat block<br>- Isopropanol<br>- Pipette tips | • 30–60% DNA recovery of input DNA. |
| In-house DNA extraction method | ~R900 for 25 reactions incl. VAT | R40 | 100 | ~3 h | | • The spin columns are homemade; the holes in the spin columns must be perfect so that there is no blockage when centrifuging the solutions. |

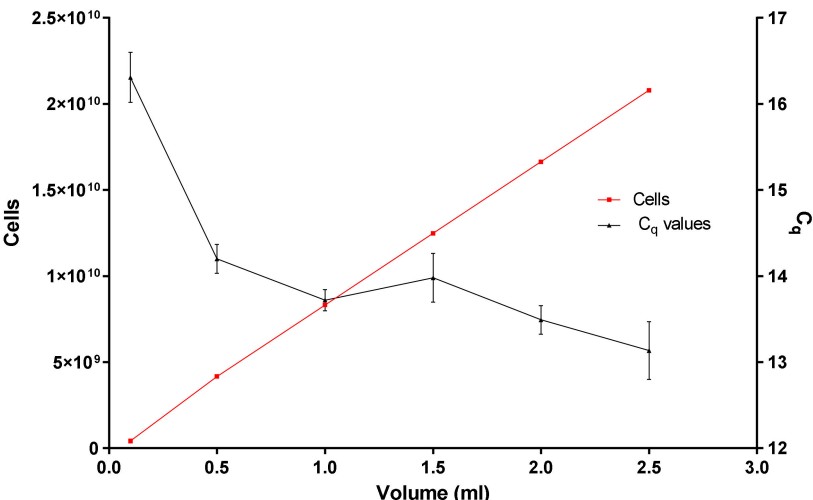

**Figure 5.** Illustrating the loading and binding capacity of the in-house DNA extraction method.

## 4. Conclusions

The first step in a molecular analysis is to extract high-quality DNA for analysis. The results of this study demonstrate the potential of the in-house DNA extraction protocol for water-testing to produce good DNA recovery, repeatability, reproducibility, and quality for PCR analysis. Additionally, the in-house DNA extraction method is also a suitable and cost-effective alternative to the available commercial DNA extraction water-testing kits. By using the in-house DNA extraction method, you can also extract genomic DNA and plasmid DNA, eliminating the need to buy separate total genomic DNA extraction and plasmid extraction mini kits. The objective in developing a cost-effective method to concentrate the bacterial cells directly from water samples followed by DNA extraction from the cells has been achieved. Using the results of this study, laboratories that cannot afford commercial water testing kits can select an appropriate DNA extraction kit for processing environmental water samples.

**Author Contributions:** Conceptualization, K.B.H. and T.G.B.; methodology, K.B.H.; software, K.B.H.; validation, K.B.H. and T.G.B.; formal analysis, K.B.H.; investigation, K.B.H.; resources, T.G.B.; data curation, K.B.H.; writing—original draft preparation, K.B.H.; writing—review and editing, T.G.B.; visualization, K.B.H. and T.G.B.; supervision, T.G.B.; project administration, K.B.H.; funding acquisition, T.G.B. All authors have read and agreed to the published version of the manuscript.

**Funding:** This research was funded by The National Research Foundation grant awarded to KBH (Grant no. 132727).

**Institutional Review Board Statement:** Not applicable.

**Informed Consent Statement:** Not applicable.

**Data Availability Statement:** Available at University of Johannesburg data repository, link https://figshare.com/s/e0acad2b6e5d1b7a8336, accessed on 2 August 2022.

**Acknowledgments:** Piet Bekker from the Medical Research Council (MRC) for the statistical analysis. J. van Staden from University of Johannesburg STATKON.

**Conflicts of Interest:** The authors declare no conflict of interest. The funders had no role in the design of the study; in the collection, analyses, or interpretation of data; in the writing of the manuscript, or in the decision to publish the results.

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
