# Peer review of "Comparison of DNA Extraction Methods for the Direct Quantification of Bacteria from Water Using Quantitative Real-Time PCR"

_water, doi:10.3390/w14223736_

Round 1
Reviewer 1 Report
The paper, “Comparison of DNA extraction methods for direct quantification of bacteria from water using quantitative real-time PCR” describes work in which the authors used commercially-available kits to extract bacterial DNA from water and compared these results to an in-house protocol. This is interesting work and would undoubtedly be of interest to the readers of Water, especially those who are working in areas where commercially-available kits are difficult to purchase due to funding or delivery issues.
The work described in this paper is very preliminary. An extremely limited number of extractions have been performed and normally I would advise against publication of this work. However, I think that because the results of this work are so promising, these limitations can be overcome with addition of text informing the reader that this is preliminary work that requires far more confirmation to ensure that the results are reproducible over time and over different samples (e.g., different water qualities). The authors touch on this on Page 3 (Lines 109-110) of the manuscript, but this is SUCH an important point; it needs to be stressed further. Another point related to this is that the number of repeats is so low that simple pipetting errors could have skewed the results of this experiment greatly – this is, in fact, likely – given the low volume of the qPCR reactions. To address this potentially major limitation of this work, I strongly feel that this wording MUST be added – into the title, into the Abstract, and into the Introduction.
A second addition that will be needed before the paper can be published: there is no measure of extraction efficiency. The authors may be assuming that the amount of cellular material in each of the samples to be extracted was equal, and that is probably not the case – thus, a higher volume of DNA in the extraction does not necessarily equate to a greater efficiency of extraction. To calculate this efficiency, extraneous standard DNA (e.g., salmon sperm DNA) must be added to each sample and quantified in the final extract. The authors MUST also recognize this limitation in the paper.
Other Specific Comments:
- There are minor grammatical edits required: 1) Line 28, change to “include”; 2) Line 31, change to “require”; 3) Lines 43-44, this is an incomplete sentence; 4) Line 65, change to “key issues”
- Lines 55-56, this observation of a “small amount” of chemicals is relative. I suggest changing the wording to: “..using relatively small amounts.”
- Line 169-170, the sentence ending in “pathogenic” requires a reference.
- Table 3, the meaning of “Input DNA vs. Calc. DNA” is unclear
- Line 232, what is meant by “remove various laboratories”? This is unclear.
- Page 3? (the page numbers started over, and line numbers ceased, not sure why), in the first paragraph, the authors describe the cost as “cheapest.” A more technical term would be “least expensive.”
Author Response
The work described in this paper is very preliminary. An extremely limited number of extractions have been performed and normally I would advise against publication of this work. However, I think that because the results of this work are so promising, these limitations can be overcome with addition of text informing the reader that this is preliminary work that requires far more confirmation to ensure that the results are reproducible over time and over different samples (e.g., different water qualities). The authors touch on this on Page 3 (Lines 109-110) of the manuscript, but this is SUCH an important point; it needs to be stressed further. Another point related to this is that the number of repeats is so low that simple pipetting errors could have skewed the results of this experiment greatly – this is, in fact, likely – given the low volume of the qPCR reactions. To address this potentially major limitation of this work, I strongly feel that this wording MUST be added – into the title, into the Abstract, and into the Introduction.
Author – included in the abstract. Line 20.
A second addition that will be needed before the paper can be published: there is no measure of extraction efficiency. The authors may be assuming that the amount of cellular material in each of the samples to be extracted was equal, and that is probably not the case – thus, a higher volume of DNA in the extraction does not necessarily equate to a greater efficiency of extraction. To calculate this efficiency, extraneous standard DNA (e.g., salmon sperm DNA) must be added to each sample and quantified in the final extract. The authors MUST also recognize this limitation in the paper.
Author- Included a statement ‘To note a limiting factor is that and extraneous DNA e.g., salmon sperm DNA was not included to each sample in the extraction process to accurately measure the extraction efficiency.’ In line 200.
Other Specific Comments:
- There are minor grammatical edits required: 1) Line 28, change to “include”; 2) Line 31, change to “require”; 3) Lines 43-44, this is an incomplete sentence; 4) Line 65, change to “key issues”
Author- Amended as indicated.
- Lines 55-56, this observation of a “small amount” of chemicals is relative. I suggest changing the wording to: “..using relatively small amounts.”
Author- amended.
- Line 169-170, the sentence ending in “pathogenic” requires a reference.
Author- included reference
- Table 3, the meaning of “Input DNA vs. Calc. DNA” is unclear
Author- ‘Input DNA’- DNA concentration quantified in ng/µâ„“ using the QubitTM fluorometer and ‘calc. DNA’-is calculated DNA from the Corbett Research Thermal Cycler software.
- Line 232, what is meant by “remove various laboratories”? This is unclear.
Author – this is one of variables used in the Z score calculations. This variable was substituted with the optimised DNA extraction method and the commercial water-testing kits.
- Page 3? (the page numbers started over, and line numbers ceased, not sure why), in the first paragraph, the authors describe the cost as “cheapest.” A more technical term would be “least expensive.”
Author - amended
The work described in this paper is very preliminary. An extremely limited number of extractions have been performed and normally I would advise against publication of this work. However, I think that because the results of this work are so promising, these limitations can be overcome with addition of text informing the reader that this is preliminary work that requires far more confirmation to ensure that the results are reproducible over time and over different samples (e.g., different water qualities). The authors touch on this on Page 3 (Lines 109-110) of the manuscript, but this is SUCH an important point; it needs to be stressed further. Another point related to this is that the number of repeats is so low that simple pipetting errors could have skewed the results of this experiment greatly – this is, in fact, likely – given the low volume of the qPCR reactions. To address this potentially major limitation of this work, I strongly feel that this wording MUST be added – into the title, into the Abstract, and into the Introduction.
Author – included in the abstract. Line 20.
A second addition that will be needed before the paper can be published: there is no measure of extraction efficiency. The authors may be assuming that the amount of cellular material in each of the samples to be extracted was equal, and that is probably not the case – thus, a higher volume of DNA in the extraction does not necessarily equate to a greater efficiency of extraction. To calculate this efficiency, extraneous standard DNA (e.g., salmon sperm DNA) must be added to each sample and quantified in the final extract. The authors MUST also recognize this limitation in the paper.
Author- Included a statement ‘To note a limiting factor is that and extraneous DNA e.g., salmon sperm DNA was not included to each sample in the extraction process to accurately measure the extraction efficiency.’ In line 200.
Other Specific Comments:
- There are minor grammatical edits required: 1) Line 28, change to “include”; 2) Line 31, change to “require”; 3) Lines 43-44, this is an incomplete sentence; 4) Line 65, change to “key issues”
Author- Amended as indicated.
- Lines 55-56, this observation of a “small amount” of chemicals is relative. I suggest changing the wording to: “..using relatively small amounts.”
Author- amended.
- Line 169-170, the sentence ending in “pathogenic” requires a reference.
Author- included reference
- Table 3, the meaning of “Input DNA vs. Calc. DNA” is unclear
Author- ‘Input DNA’- DNA concentration quantified in ng/µâ„“ using the QubitTM fluorometer and ‘calc. DNA’-is calculated DNA from the Corbett Research Thermal Cycler software.
- Line 232, what is meant by “remove various laboratories”? This is unclear.
Author – this is one of variables used in the Z score calculations. This variable was substituted with the optimised DNA extraction method and the commercial water-testing kits.
- Page 3? (the page numbers started over, and line numbers ceased, not sure why), in the first paragraph, the authors describe the cost as “cheapest.” A more technical term would be “least expensive.”
Author - amended

Reviewer 2 Report
I have reviewed the manuscript entitled “Comparison of DNA extraction methods……” submitted to your esteemed journal by Hoorzook and Barnard.
A low-cost method to directly extract DNA from bacteria in water samples will avoid the need for culture and simplify pathogenic bacteria detection in water. This manuscript has attempted to develop a DNA extraction method from E. coli using an in-house developed method. They also extracted DNA using four commercial DNA extraction kits from the same bacterial species to tell that the in-house method can reduce the cost many folds. I have several comments on this paper that the authors may consider addressing to improve the manuscript's quality.
General comments:
As a methodology development paper, the readers would be interested to see the details of the DNA extraction method. The qPCR method description section is very patchy, and there is no information on amplicon size and primer specificity. I understand that the method is intended for concentrating bacteria from the water before DNA extraction. But I do not see any such efforts from actual samples (collected from water sample), but they have diluted the cultured bacteria in water. Most importantly, do they want to isolate total DNA or focus on genomic DNA or Plasmid DNA. There is no information on spin column preparation.
Specific comments:
- DNA extraction (Lines 78-118): What is the composition of lysis buffer? What did you use as a negative control? What is the composition of the wash buffer? Did you prepare these solutions by yourself or purchase them? How did you prepare the spin column?
Please provide details about the commercial kits (Catalogue number, company name,
and location). Please provide information about DNA quantity and quality.
- qPCR (Lines 119- 131): Provide the unit of Taq DNA polymerase, dNTP mix, and Mg2+. I am surprised you have used 5mM (very concentrated) of the primers. What is the accession number of the gene gasAB? What is the amplicon size?
3. Table 3: I do not understand the meaning of the lower half of the table (input DNA vs. Calc. DNA). The slope value -2.5 and -5.7 in two commercial kits suggest that PCR amplification is not good, which might be due to improper thermal protocol or the presence of inhibitors, or both. Please explain, as the PCR principle is broken here. For the Water Master kit, the Y-intercept is 36; that means there was no template.
- Did you see the amplicon on a gel?
Taken together, this paper has several deficiencies. I am unable to recommend this manuscript for publication in the current form.
Author Response
Reviewer Report 2:
Comments and Suggestions for Authors
I have reviewed the manuscript entitled “Comparison of DNA extraction methods……” submitted to your esteemed journal by Hoorzook and Barnard.
A low-cost method to directly extract DNA from bacteria in water samples will avoid the need for culture and simplify pathogenic bacteria detection in water. This manuscript has attempted to develop a DNA extraction method from E. coli using an in-house developed method. They also extracted DNA using four commercial DNA extraction kits from the same bacterial species to tell that the in-house method can reduce the cost many folds. I have several comments on this paper that the authors may consider addressing to improve the manuscript's quality.
General comments:
As a methodology development paper, the readers would be interested to see the details of the DNA extraction method. The qPCR method description section is very patchy, and there is no information on amplicon size and primer specificity. I understand that the method is intended for concentrating bacteria from the water before DNA extraction. But I do not see any such efforts from actual samples (collected from water sample), but they have diluted the cultured bacteria in water. Most importantly, do they want to isolate total DNA or focus on genomic DNA or Plasmid DNA.
Author- amplicon size and reference included in article. The method isolates both genomic and plasmid DNA.
There is no information on spin column preparation.
Author- included in methodology section.
Specific comments:
- DNA extraction (Lines 78-118): What is the composition of lysis buffer? What did you use as a negative control?
Author- amended in the article.
What is the composition of the wash buffer? Did you prepare these solutions by yourself or purchase them? How did you prepare the spin column?
Author- Included buffer preparations in Methodology section.
Please provide details about the commercial kits (Catalogue number, company name,
and location). Please provide information about DNA quantity and quality.
Author – I have added cat log number to the kits I had the manual. I will source the cat log no. and will include it.
qPCR (Lines 119- 131): Provide the unit of Taq DNA polymerase, dNTP mix, and Mg2+. I am surprised you have used 5mM (very concentrated) of the primers.
Author- included units.
- What is the accession number of the gene gadAB? What is the amplicon size?
Author – amplicon size included in the article
- Table 3: I do not understand the meaning of the lower half of the table (input DNA vs. Calc. DNA).
Author- ‘Input DNA’- DNA concentration quantified in ng/µâ„“ using the QubitTM fluorometer and ‘calc. DNA’-is calculated DNA from the Corbett Research Thermal Cycler software.
- The slope value -2.5 and -5.7 in two commercial kits suggest that PCR amplification is not good, which might be due to improper thermal protocol or the presence of inhibitors, or both. Please explain, as the PCR principle is broken here.
Author- This was tested on reference strains thus no presence of inhibitors.
- For the Water Master kit, the Y-intercept is 36; that means there was no template.
Author – I don’t understand how you deduced there is no template?
- Did you see the amplicon on a gel?
Author- Yes I have. Previous experiments I performed qualitative PCR for presence and absence of the gene. For this study we did not test for qualitative PCR i.e., PCR with agarose gels. We tested using quantitative PCR to provide values using absolute QPCR. Quantity of the DNA for each kit is provided in Table 2 and information was analysed with QPCR data. Unfortunately in terms of the quality or purity I did not do OD260/OD280 since I quantified DNA using fluorometre that only quantified the DNA.

Round 2
Reviewer 2 Report
I have reviewed the revised manuscript entitled "Comparison of DNA extraction methods……" submitted to your esteemed journal by Hoorzook and Barnard. Here are some comments for making the manuscript better.
General comment:
11. When the Cq value is very high (in your case 37), it isn't easy to quantify the DNA amount with certainty by qPCR. One may get false negative results. When DNA quantity is too low (every minute), digital PCR (ddPCR) is a better method for quantification by Poisson distribution.
22. Spin column preparation (Lines 109-157). This is possibly the most important section in this method development paper. I suggest presenting this section in a graphical workflow format, which will help the readers to follow this section easily.
Specific comments:
- Please provide the rationale for why gadAB was chosen for DNA quantification.
- Please check the primer concentration. Is it 5mM or 5µM?
- Line 150: "The extracted DNA…..reaction" is redundant.
Author Response
General comment:
- When theCq value is very high (in your case 37), it isn't easy to quantify the DNA amount with certainty by qPCR. One may get false negative results. When DNA quantity is too low (every minute), digital PCR (ddPCR) is a better method for quantification by Poisson distribution.
Author- If the Cq value is too high to accurately quantify the DNA it is recommended that the extracted DNA is diluted to obtain a lower Cq value. This will be linked to the standard curves created to determine the detection limits of the qPCR (Klymus et al 2020).
- Spin column preparation (Lines 109-157). This is possibly the most important section in this method development paper. I suggest presenting this section in a graphical workflow format, which will help the readers to follow this section easily.
Author- Inserted figure.
Specific comments:
- Please provide the rationale for why gadAB was chosen for DNA quantification.
Author- The gadAB gene codes for the Glutamate decarboxylase enzyme. The enzyme gad has been reported to be limited to E. coli and is encoded by two virtually identical genes, gadA and gadB. GadA and gadB is found in both commensal and pathogenic E. coli therefore it was chosen.
- Please check the primer concentration. Is it 5mM or 5µM?
Author- Corrected to 5µM.
- Line 150: "The extracted DNA…..reaction" is redundant.
Author- Noted and was removed.

Round 3
Reviewer 2 Report
None